# Oncolytic Efficacy of a Recombinant Vaccinia Virus Strain Expressing Bacterial Flagellin in Solid Tumor Models

**DOI:** 10.3390/v15040828

**Published:** 2023-03-24

**Authors:** Yasmin Shakiba, Pavel O. Vorobyev, Victor A. Naumenko, Dmitry V. Kochetkov, Ksenia V. Zajtseva, Marat P. Valikhov, Gaukhar M. Yusubalieva, Yana D. Gumennaya, Egor A. Emelyanov, Alevtina S. Semkina, Vladimir P. Baklaushev, Peter M. Chumakov, Anastasia V. Lipatova

**Affiliations:** 1Moscow Institute of Physics and Technology, 141701 Dolgoprudny, Russia; yasi.shakiba@phystech.edu; 2Engelhardt Institute of Molecular Biology, Russian Academy of Sciences, 119991 Moscow, Russia; 3Department of Neurobiology, V. Serbsky Federal Medical Research Centre of Psychiatry and Narcology of the Ministry of Health of the Russian Federation, 119034 Moscow, Russia; 4Department of Medical Nanobiotechnology, Pirogov Russian National Research Medical University, 117997 Moscow, Russia; 5Federal Research and Clinical Center for Specialized Types of Medical Care and Medical Technologies FMBA of Russia, 115682 Moscow, Russia

**Keywords:** vaccinia virus, LIVP, flagellin, cytokine analysis, virotherapy, IVIS

## Abstract

Oncolytic viral therapy is a promising novel approach to cancer treatment. Oncolytic viruses cause tumor regression through direct cytolysis on the one hand and recruiting and activating immune cells on the other. In this study, to enhance the antitumor efficacy of the thymidine kinase-deficient vaccinia virus (VV, Lister strain), recombinant variants encoding bacterial flagellin (subunit B) of *Vibrio vulnificus* (LIVP-FlaB-RFP), firefly luciferase (LIVP-Fluc-RFP) or red fluorescent protein (LIVP-RFP) were developed. The LIVP-FLuc-RFP strain demonstrated exceptional onco-specificity in tumor-bearing mice, detected by the in vivo imaging system (IVIS). The antitumor efficacy of these variants was explored in syngeneic murine tumor models (B16 melanoma, CT26 colon cancer and 4T1 breast cancer). After intravenous treatment with LIVP-FlaB-RFP or LIVP-RFP, all mice tumor models exhibited tumor regression, with a prolonged survival rate in comparison with the control mice. However, superior oncolytic activity was observed in the B16 melanoma models treated with LIVP-FlaB-RFP. Tumor-infiltrated lymphocytes and the cytokine analysis of the serum and tumor samples from the melanoma-xenografted mice treated with these virus variants demonstrated activation of the host’s immune response. Thus, the expression of bacterial flagellin by VV can enhance its oncolytic efficacy against immunosuppressive solid tumors.

## 1. Introduction

Oncolytic viruses (OVs) have become a promising platform for treating cancer. OVs have the intrinsic capacity to infect neoplastic cells, mediate cellular dysfunction and promote cell death, causing tumor regression [1]. During the past decade, oncolytic viruses have been genetically modified to improve their therapeutic efficacy and decrease potential side effects. Modifications include the deletion of genes responsible for pathogenicity, enhancing viral replication and lytic ability, or immunomodulation properties by expressing various cytokines or chemokines [2,3].

Vaccinia virus (VV), a member of the Poxviridae family, has several inherent characteristics that make it suitable for oncolytic virotherapy. First, its large genomes can accommodate multiple transgenes, making it easier to perform genetic manipulation. Furthermore, their genomes encode some immune-modulating functions to evade the host’s immune responses, and finally, they display an established high safety profile [4,5]. The oncolytic properties of vaccinia virus (VV) recombinant strains have been intensively studied in multiple clinical trials. For example, the granulocyte-macrophage colony-stimulating factor (GM-CSF)-expressing recombinant, JX-594, is an oncolytic vaccinia virus with increased selectivity for replication in cancer cells. It has been tested successfully in clinical trials [6,7].

Bacterial flagellins have been used as a vaccine adjuvant. Flagellins are ligands for the toll-like receptor-5 (TLR-5). TLR-5 activation has been shown to increase necrosis and cause tumor regression effectively [8,9]. In addition, bacterial flagellins strongly stimulate the innate immune response in mammalian cells by activating transcription factor NF-κB (nuclear factor kappa, enhancer of B cells), which regulates immune responses to the infection [9]. Additionally, the presence of flagellin promotes virus attachment to the epithelial cells and increases virus entry via the TLR5-dependent activation of NF-κB [10].

We decided to develop a recombinant oncolytic variant of VV, the Lister strain from the Institute of Virus Preparation (LIVP), expressing flagellin to enhance the anticancer activity. Previously, it was demonstrated that LIVP-VV is a perspective platform to create recombinant virus strains with promising oncolytic potentials [11]. In the current study, we armed LIVP with bacterial Flagellin subunit B (FlaB), the product of the flaB gene in *Vibrio vulnificus*, alongside the reporter gene tagRFP (red fluorescent protein), and evaluated its oncolytic efficacy in comparison with the LIVP expressing only tagRFP. Moreover, we developed the LIVP to express firefly luciferase (FLuc) and tagRFP for the bioluminescence imaging and investigation of the viral biodistribution in the host’s body. All variants have viral thymidine kinase (TK) gene ablation. The deletion of TK is a common strategy to enhance VV tumor selectivity [12]. The reason is that, in normal cells, the TK level peaks only during the S-phase of the cell cycle and is almost undetectable at other times, while the TK activity in cancerous cells is constantly high. Thus, viruses with the deletion of TK will develop attraction toward neoplastic cells as TK catalyzes a critical step in nucleotide synthesis and viral replication [13].

Here, we assessed the potential of the TK-deleted LIVP with or without expression of FlaB to regress B16 melanoma allografted tumors in C57BL/6 mice and 4T1 breast cancer and CT26 colorectal carcinoma tumors in BALB/c mice.

## 2. Materials and Methods

### 2.1. Cell Lines

BHK-21 (Baby Hamster Kidney), B16 murine melanoma, CT26 murine colon carcinoma, 4T1 murine breast cancer, and HEK-293T, Raw 264.7 macrophage cell lines were purchased from the American Type Culture Collection (ATCC, Manassas, VA, USA) (ATCC product numbers CCL-10, CRL-6475, CRL-2639, CRL-3406, CRL-3216, and TIB-71 respectively). Additionally, Rat-2-TK (rat kidney cells deficient in the TK expression) was provided by doctor Nadezhda Grinenko (Serbskiy Institute, Moscow, Russia). All cell lines were cultured in a complete DMEM/Glutamax (Gibco, Billings, MT, USA) supplemented with 10% fetal bovine serum (FBS) (Gibco, USA) and 100 I.U./mL penicillin and 100 μg/mL streptomycin (PanEco, Moscow, Russia). After virus infection, cell monolayers were maintained in a medium identical to the growth medium but with 2% FBS. All cells were incubated at 37 °C in a 5% CO_2_ atmosphere. Cell counting was performed with an Improved Neubauer hemocytometer.

### 2.2. Viruses

The Lister strain from the Moscow Institute of Viral Preparation (LIVP strain) of vaccinia virus [14] was used to develop the recombinant viral strains in this study. A strain encoding only the tagRFP (red fluorescent protein) reporter gene and a strain expressing the bacterial flagellin gene (FlaB) from *Vibrio vulnificus*, alongside tagRFP (LIVP-FlaB-RFP) linked by the P2A self-processing peptide in the bicistronic cassette, were constructed. Furthermore, for the in vivo imaging, the LIVP expressing firefly luciferase with the tagRFP reporter gene (LIVP-FLuc-RFP) was developed. All transgenes were inserted into the TK-locus under the control of a 7.5k promoter (Figure 1A).

Briefly, the fragment encoding FlaB was synthesized by Integrated DNA Technologies, Inc (Coralville, IA, USA). The tagRFP plasmid (Evrogen, Moscow, Russia) and firefly luciferase cDNA (Promega, Madison, WI, USA) were purchased. Specific fragments were amplified by PCR, then inserted by sticky-ends ligation into the VV recombination plasmid vector that was developed prior at the Laboratory of Cell Proliferation, Engelhardt Institute of Molecular Biology, Moscow, Russia. Recombinant strains were obtained, as was described before [15]. Shortly, the transfection of the plasmid vector was performed using Lipofectamine 3000 (Thermo Fischer, Waltham, MA, USA) in HEK-293T cells infected with wild-type LIVP. Then, recombinant TK-deleted clones were enriched using Rat-2-TK cells (rat kidney cells deficient in the TK expression) infected at MOI 1 by total viral stock from the previous step and treated with 5′-Bromo-2′-deoxyuridine. After 5 rounds of subcloning, a pure plaque was obtained. The correctness of the insertion sequences in the recombinant variants was confirmed by Sanger sequencing. Additionally, the expression of the red fluorescence is detectable in the cell cultures infected with the recombinant variants by the fluorescent microscope.

### 2.3. Virus Titration

BHK-21 cells were used for virus propagation and viral titrations. First, all the recombinant viruses were purified by sucrose gradient centrifugation, as was described before [16]. The cells were seeded in 96-well plates, 1 × 10^4^ per well. The next day, media were removed, and the cells were infected with 10-fold serial dilutions of the viruses and incubated in DMED media supplemented with 2% FBS. After 48 h, the cytopathic effect could be observed and quantified, and TCID50 (50% tissue culture infective dose) was assessed by using the Reed–Muench method [17].

### 2.4. Virus Replication Efficiency

In a 96-well plate, 1 × 10^4^ of BHK-21, B16, CT26, and 4T1 cells were seeded per well. After monolayer formation, the cells were infected with viruses at MOI 1 and 10. Seventy-two hours post-infection, the supernatants were collected and titrated by infection of fresh BHK-21 cells by 10-fold serial dilutions, as was described before [18]. The quantitative titer was assessed by using the Reed–Muench method [17].

### 2.5. Evaluation of Firefly Luciferase Expression in Vitro by Bioluminescence Assay

In order to evaluate the functional activity of the firefly luciferase expression of the recombinant virus LIVP-FLuc-RFP, BHK-21 cells were seeded on a 96-wells plate, then, infected with MOI 0.1,1, and 10 of this recombinant virus. Twenty-four hours after the infection, luciferin was added to the wells at a final concentration of 30 μg/mL and incubated for 10 minutes. Afterward, bioluminescence detection was conducted using a ClarioSTAR microplate reader (BMG, New York, NY, USA).

### 2.6. In Vitro Assessment of Flagellin Production by Western Blot Analysis

A Western blot was completed to verify the flagellin expression. Confluent BHK-21 cell monolayers in a 6-well plate were infected at MOI 1 of the LIVP-FlaB-RFP or LIVP-RFP, and the not-infected cells served as additional controls. The cell lysates were collected in RIPA buffer, supernatants were collected, and we proceeded as was described earlier [19]. In brief, equal quantities of all the samples were subjected to 12% SDS-PAGE. Proteins were transferred onto a polyvinylidene fluoride (PVDF) membrane (Millipore, Burlington, MA, USA), blocked using 4% non-fat milk and incubated for 1 h at room temperature with rabbit anti-serum containing polyclonal antibodies against FlaB diluted in 2% non-fat milk containing 0.05% Tween. The rabbit anti-serum was kindly provided by Joon Haeng Rhee’s laboratory (Chonnam National University, South Korea). We used FlaB anti-serum at a dilution of 1:6000, anti-tagRFP rabbit polyclonal at a dilution of 1:5000 (Santa-Cruz Biotechnology, Dallas, TX, USA), and anti-Actin monoclonal at a dilution of 1:2000 (Sigma, St. Louis, MO, USA). Secondary anti-rabbit antibodies were used in a dilution of 1:3500 (sc-2357, Santa-Cruz Biotechnology, Dallas, TX, USA). Finally, bands were visualized using the ChemiDoc system (Bio-Rad, Hercules, CA, USA).

### 2.7. Evaluation of Viral Cytotoxicity

A cytotoxicity assay was performed to evaluate the direct lytic activity of the new strains. The 4T1, CT26, B16, and BHK-21 (as a control) cell lines were seeded at 1 × 10^4^ cells per well in 96-well plates and infected with the MOI 10 and 1 of the LIVP-FlaB-RFP and LIVP-RFP. Cell viability was determined by MTT assay at the indicated time points of 24, 48, and 72 h post-infection [20].

### 2.8. Assessment of the Virus Kinetics *by Flow Cytometry*

The RFP expression in the infected cells correlates to viral replication. The 4T1, CT26, B16, and BHK-21 (as the reference) cells were seeded at 1 × 10^5^ cells per well in 24-well plates overnight and then infected with the LIVP-FlaB-RFP or LIVP-RFP at MOI 10 and 1. Cells were harvested 24 and 48 h after infection for a flow cytometry analysis. The samples were analyzed by detecting red fluorescence in the PE channel using a BD LSR Fortessa cytofluorimeter (Beckman Dickinson, Franklin Lakes, NJ, USA) with 10,000 events per sample in 3 technical replicates. The analysis was performed using Flowing Software 2.0 (Turku Bioscience Center, Turku, Finland).

### 2.9. Evaluation of Virus-Induced Cytotoxicity in the Presence of Macrophages in Vitro

The B16, CT26, and 4T1 cell cultures on 96-well plates, at a confluency of 1 × 10^4^ cells per well, were prepared, and serial dilutions of the recombinant virus LIVP-FlaB-RFP or LIVP-RFP were added to the cells with 4 replications for each dilution and incubated for 2 h. Media were discarded, and fresh 1% FBS-supplemented media were added to the wells. In order to carry out the co-cultivations, Raw264.7 macrophage cells were dyed with CellTracker Green CMFDA Dye (Thermo Fisher, Waltham, MA, USA), and 1 × 10^3^ of them were added to each well of the infected cell cultures on 96-well plates 24 h after viral infection. The plates were incubated, and 24, 48, and 72 h later, the MTT-viability assay [20] was performed to measure the cytotoxicity of the viruses with or without the presence of macrophages as an effector on different cell lines.

### 2.10. Mice Models and Virotherapy

Female 6-week-old C57BL/6 mice were used to establish the murine B16 melanoma models; BALB/C mice were used to establish the 4T1 breast cancer or CT26 colon carcinoma syngeneic cancer models. Furthermore, 1 × 10^6^ tumor cells were injected subcutaneously into their right hind flanks. All animals used in this study were authorized by the Engelhardt Institute of Molecular Biology, Moscow, Russia. The mice were housed under a controlled temperature and humidity, with free access to food and water.

The intravenous (IV) virotherapy was conducted one week after tumor implantation, while the tumor volumes reached approximately 50 mm^3^. The melanoma, breast cancer, and colon carcinoma mice models were divided into three groups (*n* = 11 for each group); one group received LIVP-FlaB-RFP, another group received LIVP-RFP, and the last group, as the control, received PBS (phosphate buffered saline). Furthermore, 5 × 10^6^ PFU of the viruses (LIVP-RFP or LIVP-FlaB-RFP) in 100 µL of PBS were injected IV two times with three-day intervals into each mouse in the corresponding groups.

The tumor diameters were measured every 2 days, and the tumor volume was calculated using the equation: V  =  ½ a^2^  ×  b, where a is the smaller of the two orthogonal measurements [21]. The mice were euthanized when the tumors reached 2000 mm^3^ in size.

### 2.11. Biodistribution of the Virus Detected by Bioluminescence Imaging in Vivo

Syngeneic tumors were established by s.c. injection of murine melanoma B16, 4T1 breast cancer, and CT26 colon carcinoma cells (2 × 10^6^ cells) in 50-µL PBS into the right hind flank of C57BL/6 and BALB/c mice, respectively. On day 5, post-implantation, the mice were treated with a single intravenous injection of VV-FLuc-RFP (1 × 10^8^ PFU in 100 µL of PBS; *n* = 8); untreated mice were used as the control (*n* = 8). For the bioluminescence imaging, mice infected with VV-FLuc-RFP were injected i.p with 150 mg/kg of firefly D-luciferin (Perkin Elmer, Waltham, MA, USA) in PBS and allowed to rest for 10 min. Imaging was conducted using the in vivo imaging system (IVIS) Spectrum CT (Perkin Elmer, USA) and the photon emission values were calculated with the Living Image 4.3 software.

### 2.12. Viral Biodistribution Quantified by qPCR

After the virus injection, tumors and different organs from the treated mice were collected. Total DNA was extracted from the tumor tissues using the DNeasy 96 Blood & Tissue Kit (Qiagen, Manchester, UK) according to the manufacturer’s instructions. The normalization of the DNA concentrations and standards was prepared as described before [22]. Quantitative PCR proceeded using 5X qPCRmix-HS SYBR (Evrogen, Moscow, Russia) on the CFX-96 qPCR system (Bio-Rad, Hercules, CA, USA). The primers for the analysis were produced by Evrogen (Moscow, Russia). They were designed on the vaccinia A21L gene (forward: 5′-CGTAAACTACAAACGTCTAAACAAGAA-3′ and reverse: 5′-CCTGGTATATCGTCTCTATCTTTATCA C-3′).

### 2.13. Histology

The histological study was performed on 3 mice from each group. The samples from tumors of treated and control mice were harvested and post-fixed in the 4% buffered formalin solution (Thermo Fisher Scientific, USA) at 4 °С overnight, followed by paraffin embedding. Furthermore, 5 µm-thick paraffin sections were prepared and stained using standard hematoxylin and eosin (H&E) staining [23].

### 2.14. Cytokine Analysis of the Sera and Tumor Tissues

To analyze the level of cytokines, ten days after the complete injections of the viruses or PBS to the melanoma mice models, sera were obtained from 3 mice from both the control and treated groups. Tumor tissues were collected, and 0.25 g of them were homogenized and processed for this experiment, as well [24]. The LEGENDplex MU Cytokine Release Syndrome Panel (13-plex) w/FP kit (BioLegend, San Diego, CA, USA) was used to determine the level of the interferons (IFN)-α and IFN-γ, the interleukins (IL)-4, IL-6, and IL-10, the chemokine C-C motif ligand 2 (CCL2 or MCP-1), CCL3 (or MIP-1α), CCL4 (or MIP-1β), the chemokine C-X-C motif ligand 9 (CXCL9 or MIG), CXCL10 (or IP-10), and the tumor necrosis factor (TNF)-α, vascular endothelial growth factor (VEGF), and granulocyte-macrophage colony-stimulating factor (GM-CSF).

### 2.15. Tumor Infiltrating Lymphocytes/Macrophages

The mice treated with the viruses or PBS were sacrificed (*n* = 6), and their tumors were collected in RPMI-1640 containing 2% FBS and 1 mg/mL collagenase IV (Sigma, St. Louis, MO, USA). After 30 minutes of incubation at 37 °C, the samples were homogenized in ice-cold PBS containing 2% FBS and 1 mM EDTA. After obtaining single cell suspension, 10^7^ of the resulting cells were incubated with Fc Block (anti-mouse CD16/CD32; 101302), followed by staining with the PE anti-mouse CD4 antibodies (100512), Pacific Blue anti-mouse CD8a (100725), PE anti-mouse F4/80 (123110) antibodies (all from BioLegend, San Diego, CA, USA). The stained cells were acquired on the BD LSR Fortessa cytofluorimeter (Beckman Dickinson, Franklin Lakes, NJ, USA), and data were analyzed using Flowing Software 2.0 (Turku Bioscience Center, Turku, Finland).

### 2.16. Statistical Analysis

All data are presented as mean ± SD. Statistical analyses were performed using unpaired *t*-tests and a two-way ANOVA analysis. Differences were considered significant where * *p* < 0.05, ** *p* < 0.01, *** *p* < 0.001, and **** *p* <0.0001. GraphPad Prism version 8.0.2 (GraphPad Software, Inc., San Diego, CA, USA) prepared all graphs and statistical analyses. 

## 3. Results

### 3.1. Construction of the Recombinant Viruses and Confirmation of Transgene Expression

The recombinant viruses were constructed using tagRFP as the selecting gene. The viruses LIVP-FlaB-RFP and LIVP-FLuc-RFP contained the insertion of FlaB and FLuc, respectively (Figure 1A). The infection of the BHK-21 cells with LIVP-FlaB-RFP and LIVP-Fluc-RFP shows strong fluorescence signals from the infected cells in both cases and the control, LIVP-RFP (Figure 1B, Appendix A). The efficient expression of luciferase in the LIVP-Fluc-RFP infected cells was assessed by bioluminescence assay, as was described (Appendix A).

Cell lysates from the infected BHK-21 cells with LIVP-FlaB-RFP were obtained and analyzed for the presence of FlaB by Western blotting assay. Additionally, non-infected cells served as the control (Figure 1D, Appendix A).

### 3.2. Assessment of Viral Recombinant Strain’s Replication Efficiency, Kinetics, and Cytotoxicity

The virus replication efficiency was measured by titration and the determination of TCID50 of the supernatant that was obtained from different infected cell cultures, including BHK-21, B16, CT26, and 4T1 by MOI 1 of LIVP-RFP or LIVP-FlaB-RFP. It appears that the presence of the transgene FlaB in the genome reduces the replication level to some extent in comparison with LIVP-RFP (Figure 2). Additionally, in this test, it has been shown that the recombinant viruses have a higher ability to reproduce in the B16 cell culture rather than in 4T1 or CT26 cell models (Figure 2).

The viral replication rate was determined by flow cytometry, which detected red fluorescent signals emitted from the infected tumor cells. BHK-21 was infected as a reference. The results imply that recombinant viruses can infect and replicate in cancer cell cultures, especially at the higher MOI (MOI 10). No significant difference was observed between LIVP-RFP and LIVP-FlaB-RFP’s ability to replicate in tumor cell lines (Figure 3).

The cytotoxicity of the viruses was assessed at 24, 48, and 72 h after infection of the different cell lines of B16, CT26, and 4T1. BHK-21 cell cultures were used as the control. Our data suggest that LIVP-FlaB-RFP has a higher level of cytotoxicity than LIVP-RFP at 10 MOI in all tested tumor cell lines, while in the lower MOI, no significant differences were observed (Figure 4).

### 3.3. Flagellin Expression Enhances Macrophages’ Cytolytic Effect in Vitro

The innate immune system recognizes bacterial flagellin by the Toll-like receptor 5 and NAIP5/NLRC4 inflammasome receptors, which are expressed mainly on macrophages, leading to a pro-inflammatory program of cell death [25]. Activated macrophages can significantly synergize the cytotoxic effect of the oncolytic virus. For modeling macrophage activation and its effects on cytotoxicity in vitro, we co-cultured RAW264.6 cells with tumor cells (B16, CT26, and 4T1 cells), pre-infected with 0.1 MOI of LIVP-RFP or LIVP-FlaB-RFP. At this MOI, the cytotoxicity of the viruses to the tumor cell lines is very low; thus, the changes by the effector (macrophage) can be easily observed (Figure 5A–C). The cytolytic effect was assessed after 48 h of co-incubation using the MTT test. The results showed that B16 cell viability decreases dramatically in the presence of LIVP-FlaB-RFP. The cytolytic effect was more significant in the co-cultured LIVP-FlaB-RFP-infected cells with macrophages (Figure 5D). The co-incubation with macrophages with the LIVP-FlaB-RFP-infected CT26 cells showed a tendency for survival decline, but the difference was not significant. In the case of the 4T1 cells, neither LIVP-FlaB-RFP nor the co-incubation of LIVP-FlaB-RFP with macrophages affects cell viability.

### 3.4. The Administration of the Recombinant VV Results in Significant Viral Deposition within Tumor with Evidence of Viral Replication in Tumor, but Not Normal Organs

For further investigation and understanding of the biodistribution of the viruses in the host’s body, we developed LIVP-FLuc-RFP. We detected the virus after injection into the C57BL/6 mice bearing the B16 melanoma tumors. At first, the virus was seen in the spleen (up to 6 H) (Appendix A). Then, from 24 h until 168 h post-infection, luciferase expression only was detected at the tumor sites, and no signal in any other organs was observed, which indicates the safety and tumor-targeted property of the recombinant LIVP virus with the long-term replication time (Figure 6). Furthermore, 24 and 168 h after virus injection, four mice were sacrificed to collect their tumor, spleen, liver, brain, ovaries, heart, kidneys, and blood for quantification of the virus by qPCR (Appendix A). Additionally, 24 h post-injection, an average of 10^5^ copies of viral genomic DNA in the tumor was recovered, which was significantly higher than the virus being detected in the spleen, ovaries, liver, and blood. Furthermore, 168 hours post-injection, the systemic virus was cleared while persisting within the tumor.

### 3.5. Flagellin Expression Enhances the Antitumor Activity of the Vaccinia Virus in the Murine Melanoma Model

In the seventh and tenth days after tumor inoculation, purified viruses were injected intravenously (Figure 7A). The tumor volume was assessed (Figure 7B), and the survival rate was determined by using the Kaplan–Meier method (Figure 7C). In all treated groups, lower tumor volumes were observed compared to the control groups, especially in the treated groups receiving LIVP-FlaB-RFP; however, it appears that the B16 melanoma model is more responsive to the treatment than the colon carcinoma or breast cancer models. Regression in the melanoma syngeneic mice models receiving LIVP-FlaB-RFP was significantly higher than the group that received only LIVP-RFP or the control. The survival rates were also significantly prolonged in all treated groups, and for the melanoma models that received LIVP-FlaB-RFP, survival was remarkably longer than in any other groups. Measurements continued until 50% of the animals in the control group reached the maximum tumor volume, which is indicated by the animal care guidelines. (data were obtained for up to 35 days).

### 3.6. Flagellin Expressing VV Modulates the Tumor Microenvironment

We measured the levels of 13 cytokines, including IFN -α, IFN-γ, IL-4, IL-6, IL-10, CCL2, CCL3, CCL4, CXCL9, CXCL10, TNF-α, VEGF, and GM-CSF from both the sera and tumor samples from the B16 melanoma mice group (*n* = 3), which had the best response to the viral therapy in comparison with 4T1 breast cancer or CT26 colon carcinoma. The CCL4 level did not change in the serum or tumor samples, and the IL-10 level did not change in the samples from the serum. In the sera samples from the treated group by LIVP-RFP, an increased level of cytokines was observed (compared with LIVP-FlaB-RFP in the treated group or the control) (Figure 8B), while in the tumor samples of the same group, this increase was not seen (Figure 8A). On the contrary, the treated group by LIVP-FlaB-RFP had a surge of cytokines in their tumor samples and not in the serum samples. (Appendix A). Antibody staining of the tumor-infiltrated lymphocytes indicated a significant increase in the levels of the CD8+ lymphocytes and macrophages treated by the oncolytic viruses, especially LIVP-FlaB-RFP (Figure 9). The histological analysis demonstrated decreased melanin deposition after therapy for the oncolytic viruses and a pronounced inflammatory infiltration and acellular foci of necrosis in tumor samples treated with LIVP-FlaB-RFP (Appendix A).

## 4. Discussion

During the past decade, genetic engineering has been employed in the oncolytic viral therapy field to increase the onco-selectivity of the viruses, including the vaccinia virus and to enhance the immune response of the host to gain antitumor immunity and prominent regression in tumors. For instance, the deletion of the thymidine kinase gene has been used to increase cancer cell selectivity [12,26]. We made a recombinant vaccinia virus from the strain LIVP, in which the thymidine kinase gene was ablated by inserting the reporter gene tag RFP to increase virus selectivity for tumor cells. Furthermore, in order to enhance the stimulation of the immune response and broaden the cytotoxicity of the virus, we constructed the variant that expresses bacterial flagellin subunit B (LIVP-FlaB-RFP).

At first, we evaluated the ability of the virus to efficiently replicate and lyse various tumor cell lines in vitro, as direct cytolysis by a virus is a main step to tumor clearance. Viruses had more potency to replicate in the B16 cell line, and transgene FlaB in the virus backbone has reduced the replication efficacy to some degree. However, on the other hand, it increased cytotoxicity. Infection of the co-culture of the B16 cell line with macrophages by LIVP-FlaB-RFP resulted in significantly increased cytolytic effects, probably due to the activation of macrophages in the presence of flagellin [27].

Moreover, in an attempt to demonstrate the recombinant virus’ onco-selectivity, we performed IVIS imaging after injection of LIVP-Fluc-RFP to the melanoma mice models; only up to 6 h after the injection signals were observed in the spleen of the mice, and afterward, up to 168 h post-infection signals were only detected in the tumor area and not in any other organs. Detection of the viral genomic DNA in the tumor and different organs by qPCR analysis confirmed the IVIS results, indicating favorable biodistribution of this recombinant virus.

The addition of the flagellin (FlaB) transgene in the TK-deleted virus backbone resulted in the most effective oncolytic strain in our in vivo studies, which significantly reduced tumor volumes, lowered tumor progression, and prolonged the life span of the cancer models, especially in B16 melanoma allografted C57BL/6 mice. Nevertheless, how the presence of FlaB contributes to tumor regression is unclear. Research has shown flagellin triggers the activation of toll-like receptor 5 (TLR-5). Subsequently, the release of cytokines occurs, which can cause necrosis in tumors [8]. Furthermore, bacterial flagellins are essential targets for the detector molecules involved in immunosurveillance. For instance, the detection of flagellin by the Nod-like receptor NCLR4 triggers activation of the ice protease-activating factor (IPAF) inflammasome that initiates caspase-1 and maturation of the interleukin (IL)-1β [28,29]. IL-1β expressed in tumors can induce macrophage chemotaxis and increase macrophage recruitment. This was demonstrated in human metastatic melanoma models in vitro [30]. Additionally, Nguyen et al. showed that the administration of flagellin B (FlaB) into TC-1 carcinoma cancer models significantly potentiated the antitumor immune responses and prolonged survival. Furthermore, FlaB increased antigen-specific IFN-γ production of the CD8^+^ T cells from the lymph nodes and spleen [31]. In another study, it was demonstrated that an injection of soluble flagellin C (FliC) into human colorectal tumors, xenografted in nude mice, significantly reduced tumor volume [32]. Conrad et al. constructed a recombinant tanapox virus armed with FliC (from *Salmonella enterica*) that exhibited excellent oncolytic activity in colorectal cancer models in comparison with other recombinant strains expressing GM-CSF or CCL2 [33].

In our investigation, the melanoma models responded to the viral therapy better than the breast cancer or colon carcinoma models. This was particularly evident in the group that received LIVE-FlaB-RFP, which is consistent with the in vitro studies that showed a higher expression level and more cytotoxicity of this virus in the B16 melanoma cell culture rather than the CT26 colon carcinoma or 4T1 breast cancer cell culture. An evaluation of the immune response to the virotherapy by LIVP-FlaB-RFP indicated significantly enhanced recruitment of the macrophages and CD8^+^ lymphocytes to the tumor site compared to the group treated by LIVP-RFP or the control. Furthermore, a cytokine analysis of the serum and tumor samples from the mice bearing B16 melanoma, treated by LIVP-FlaB-RFP, indicated a significant elevation of the cytokines: TNF-α, GM-CSF, CXCL9, IL-4, and, to a lower extent, the elevation of IL-10, CCL3, CXCL10, IFN-α, IFN-γ, VEGF, and CCL2 in samples from the tumors treated by LIVP-FlaB-RFP, while in the serum analysis of this group, a slight increase of only TNF-α, VEGF, and IL-6 was observed. In the treated group, the LIVP-RFP cytokine analysis suggested elevation of CXCL10 in the tumor samples and IFN-α, IFN-γ, CCL2, CCL3, CXCL9, and CXCL10 in the sera samples. This data indicates that the virus expressing flagellin has more tropism toward the tumor and has more potential to penetrate the tumor microenvironment (TME), which leads to localized cytokine promotion at the tumor site, which suggests a higher rate of immune cell infiltration and overcoming immunosuppressed TME, which can profoundly turn the cold tumor into a hot status. ‘Hot’ tumors are characterized by the accumulation of cytokines and T-cell infiltration and have a better response to immunotherapy [34]. Melanoma could be considered the “hot” tumor among the models used in our research; therefore, flagellin expression in the infected cells readily activates the immune cell. In other cases, the effect of flagellin is not so drastic. It could be improved if flagellin is expressed in a secreted form.

GM-CSF (granulocyte-macrophage colony-stimulating factor) is a potent cytokine that promotes the differentiation of myeloid and dendritic cells, which is responsible for presenting tumor antigens for the priming of antitumor cytotoxic T-lymphocytes and is capable of the regulation of immunosuppressive TME. In oncolytic viral therapy, this cytokine has shown significant tumor regression in colorectal and hepatic adenocarcinomas [35]. In phase II clinical trials, a recombinant oncolytic herpes simplex virus type-1 encoding GM-CSF resulted in the regression of unresectable stage III or IV melanomas. Based on this result, a phase III clinical trial in patients with unresectable Stage III b or c and Stage IV melanoma was initiated and had successful outcomes [36].

CXCL9 (C-X-C motif chemokine ligand 9) has a conserved protein function between mice and humans. It has been reported to be associated with promoted intra-tumor infiltration of immune cells, T-cell accumulation, recruitment of activated CD8^+^ cytotoxic T and CD4^+^ T-helper cells, delayed ascites formation, and improved survival [37]. Seitz et al. declared the overexpression of CXCL9 inhibits tumor growth in ovarian cancer in immune checkpoint blockade-resistant mice [38].

TNF-α (tumor necrosis factor-alpha) has a paradoxical effect in cancer therapy; depending on the cellular and cytokines context, it induces either cancer cell death or survival. Although TNF-α shows potent antitumor activity in various animal cancer models, due to its toxicity, clinical use of TNF-α is now limited to isolated limb perfusion (ILP) for melanoma and soft tissue sarcoma. Recently, it has been described that systemic administration of TNF-expressing tumor cells can reduce tumor growth and metastatic colonies in immunocompetent mice [39,40]. Moreover, a recombinant variant of adenovirus expressing TNF-α displayed cancer-eradicating potency by elevated apoptosis and necrosis with an enhanced level of cytotoxic T cells [41].

The cytokines discussed above have significantly escalated in treated melanoma models with LIVP-FlaB-RFP. We assume the surge of these cytokines can contribute to a lower tumor volume and extended lifespan of treated mice. This recombinant variant was able to drive a substantial increase in the GM-CSF, CXCL9, and TNF-α chemokine gradient between the tumor and blood. Additionally, a rise in other cytokines, including IFN-α, IFN-γ, IL-10, CCL3, and CXCL10, was seen in both melanoma-treated groups receiving LIVP-RFP or LIVP-FlaB-RFP. Nonetheless, the increase was not statistically significant, but various research revealed that these cytokines could play an essential role in tumor regression [42,43].

One of the significant disadvantages of viral immunotherapy is Cytokine Release Syndrome (CRS), typically associated with the surge of IFN-γ, IL-2, IL-4, IL-6, IL-8, etc. [44]. Additionally, our cytokine analysis determined an uplifted level of inflammatory cytokines, such as VEGF, IL-6, and CCL2, that can cause the progression of the tumor and initiate CRS, which limits the treatment and has a negative impact on the outcome. It is worth mentioning that inflammatory cytokine IL-4 was higher in the control group than in the treated groups in the sera analysis. Nevertheless, targeting and inhibiting the pathways that lead to the production or function of these inflammatory cytokines by anti-cytokine antibodies is a novel approach to cancer therapy, which can prevent CRS problems without compromising antitumor activity [45,46]. We hypothesize that combining these recombinant variants with immune checkpoint blockades or antibodies targeting inflammatory cytokines can optimize the treatment process. 

## 5. Conclusions

Our study represents a simple approach to exploiting immune-suppressive tumors and converting them into opportunities for targeted immunotherapy. Our results support the notion that VV-LIVP is a prospective oncolytic strain, and arming it with FlaB has the potential to be an effective oncolytic virus, mediating its oncolytic efficacy via direct cytolysis and activation of the immune response at the TME. However, further research should be performed due to an imperfect understanding of flagellin’s mechanism of action. We presume that administering this recombinant variant with other anticancer medications would improve treatment outcomes.

## Figures and Tables

**Figure 1 viruses-15-00828-f001:**
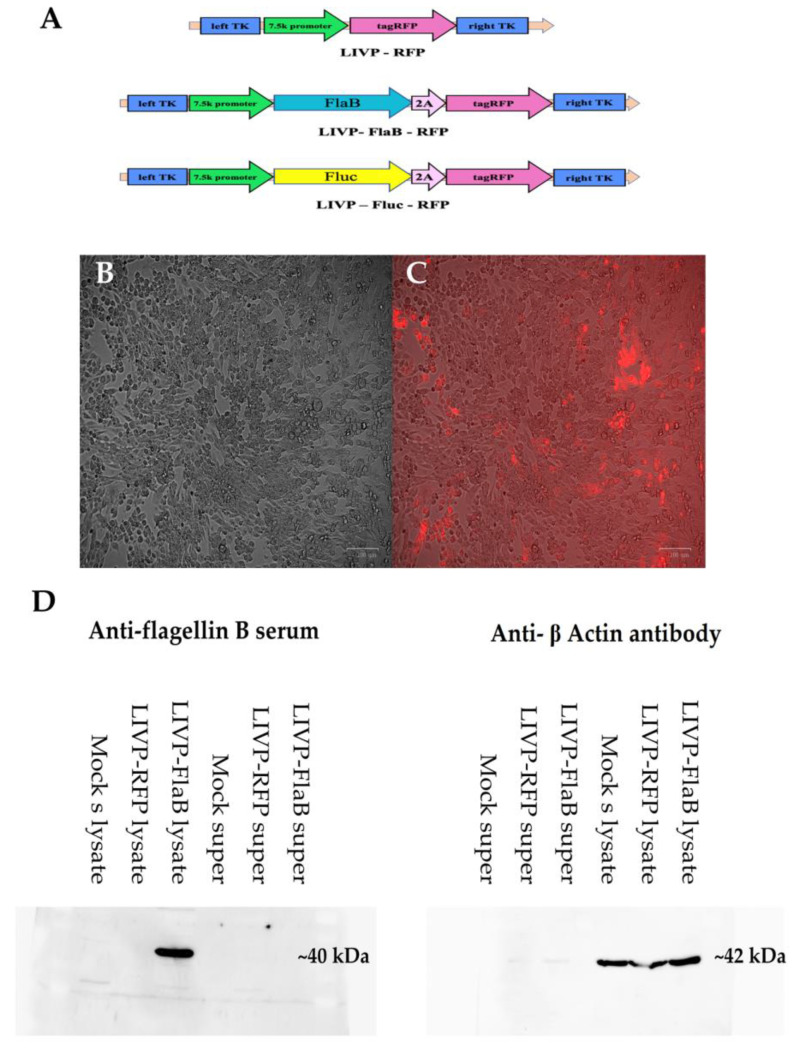
Characteristics of recombinant viruses. Schematic illustration of the plasmid’s constructions (**A**). Microphotographs of BHK-21 cells 24 h after infection with MOI 1 LIVP-FlaB-RFP (20× magnification, scale bar represents 100 µm), pictures are taken on a red fluorescent channel (**B**) and bright field microscope (**C**), (microphotographs of BHK-21 cells infected with LIVP-RFP and LIVP-Fluc-FRF are presented in Appendix A). (**D**) Western blotting analysis of FlaB (~40 kDa) in cell lysate/supernatant (super) of infected cells by LIVP-FlaB-RFP, LIVP-RFP, and non-infected cells (control); the anti-β-Actin antibody was used as a control. Cell lysate of the infected cells by LIVP-FlaB-RFP is positive for FlaB.

**Figure 2 viruses-15-00828-f002:**
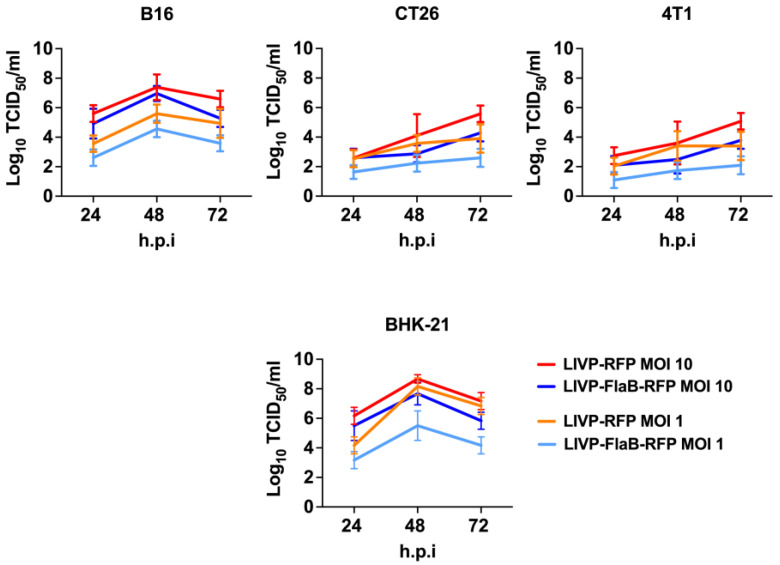
Viral replication efficiency. Comparison of replication efficiency of the recombinant viruses with or without expression of the flagellin transgene in various cell cultures.

**Figure 3 viruses-15-00828-f003:**
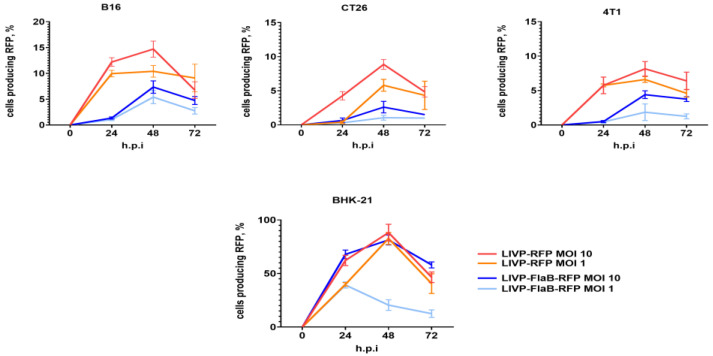
Viral kinetics. Flow cytometry assessment of virus replication kinetics at 24, 48, and 72 h post-infection (h.p.i) in the different cell cultures of BHK-21, B16, CT26, and 4T1 with recombinant virus variants.

**Figure 4 viruses-15-00828-f004:**
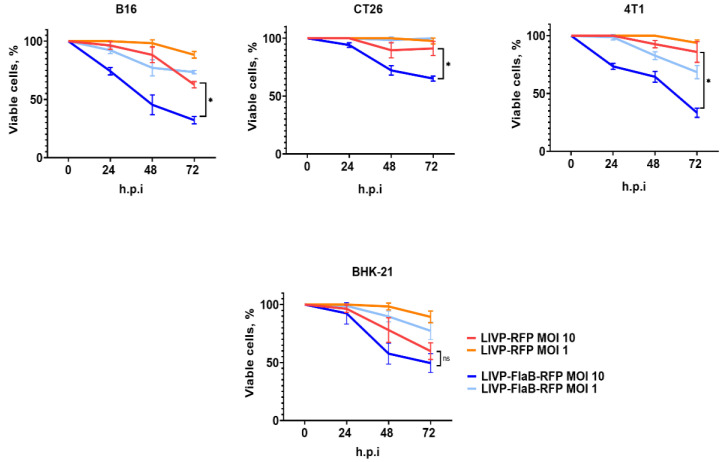
Virus cytotoxicity. Percentage of viable cells up to 72 h after infection (h.p.i), evaluated by MTT assay, in BHK-21, B16 murine melanoma, CT26 murine colon carcinoma, 4T1 breast cancer cell lines. ANOVA was performed for statistical analysis; * *p* < 0.05 indicates significance.

**Figure 5 viruses-15-00828-f005:**
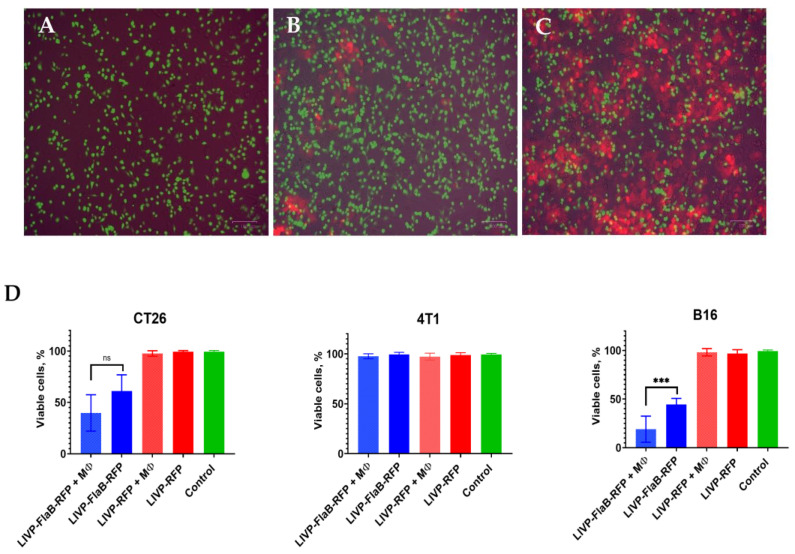
Macrophage cytolytic effect. Microscopic pictures (20× magnification, scale bar represents 100 µm) of the B16 cells co-cultivated with RAW264.7, pre-stained with CellTracker Green (**A**) and infected with 0.1 MOI of LIVP-FlaB-RFP (**B**) or LIVP-RFP (**C**). (**D**) Qualitative assessment of cell viability using MTT assay at 48 h post-infection of co-cultivation, normalized to non-infected co-cultured cells as a control. For statistical analysis, an ANOVA test was performed; *** *p* < 0.001 indicates significance.

**Figure 6 viruses-15-00828-f006:**
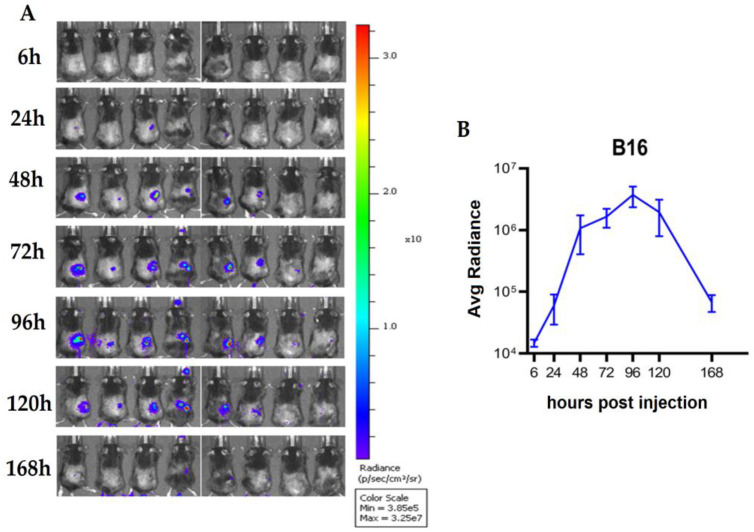
In vivo imaging. Virus detection by IVIS through time after injection of LIVP-FLuc-RFP to the B16 syngeneic mice. (**A**) Bioluminescence imaging of mice bearing B16 melanoma tumors. (**B**) Average radiance after virus injection; bioluminescence was detected up to 168 h after virus injection, and the maximum expression of luciferase was observed 96 h after infection.

**Figure 7 viruses-15-00828-f007:**
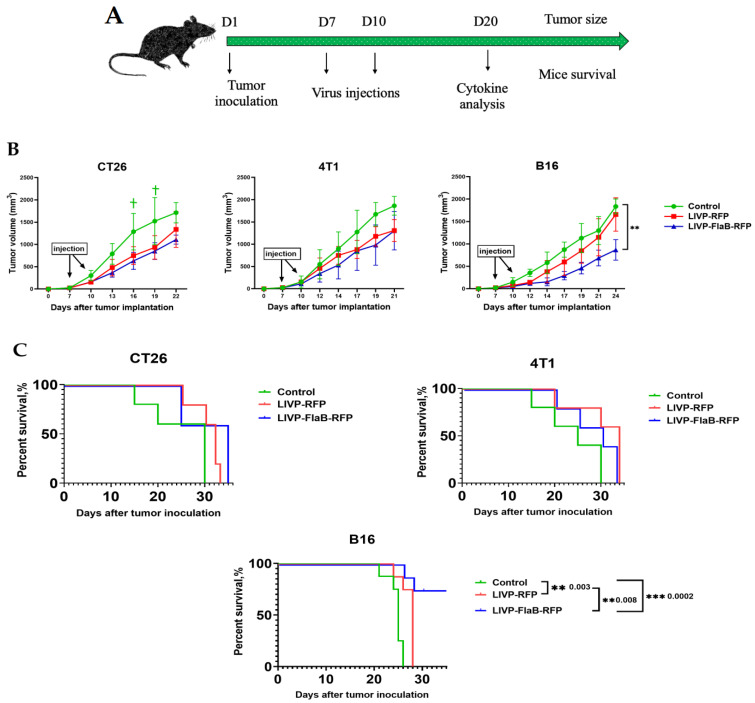
Virotherapy. (**A**) Schematic of the treatment protocol mice (*n* = 11) were injected subcutaneously with B16, CT26, or 4T1 cells on day 1. Seven days after tumor inoculation, viruses were injected intravenously. Then, the mice received a booster shot on day 10. Ten days after treatment, three mice from each group were sacrificed to obtain sera and tumor samples for cytokine analysis. (**B**) Tumor progression (B16, 4T1, CT26) in treated groups with LIVP-RFP or LIVP-FlaB-RFP compared to the control groups. The symbol ✝ indicates the animal’s sacrifice in control groups. (**C**) Overall survival percentages of treated and control mice based on the Kaplan–Meier method. For statistical analysis, ANOVA and survival analysis were performed; ** *p* < 0.01 and *** *p <* 0.001 indicate significance.

**Figure 8 viruses-15-00828-f008:**
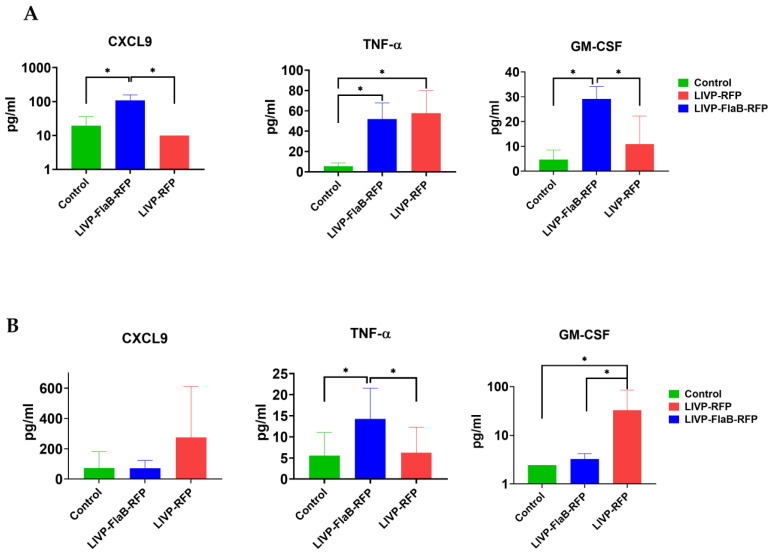
Cytokine analysis. (**A**) Cytokine analysis of the tumor-infiltrated fluids (TIFs) and (**B**) serum samples from the B16 melanoma models treated by LIVP-RFP or LIVP-FlaB-RFP compared with the untreated mice (control group). For statistical analysis, ANOVA was performed; * *p* < 0.05 indicates significance.

**Figure 9 viruses-15-00828-f009:**
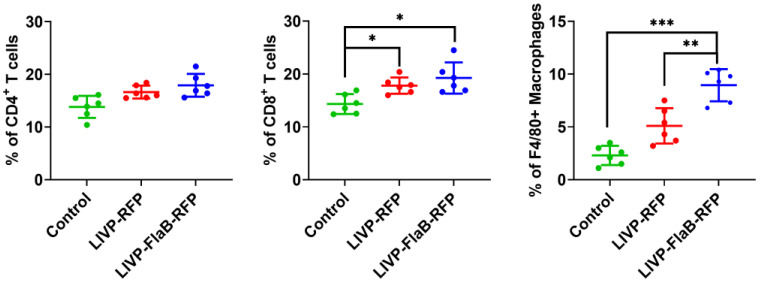
**Analysis of tumor infiltrating lymphocytes/macrophages.** B16 melanoma tumor samples were collected 72 hours after treatment to evaluate the levels of F4/80^+^, CD4^+^, and CD8^+^ cells. For statistical analysis, ANOVA was performed; * *p* < 0.05, ** *p* < 0.01; *** *p* < 0.001 indicates significance.

## Data Availability

The data analyzed in this current research are available from the corresponding author on request.

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
