# Peer review of "Oncolytic Efficacy of a Recombinant Vaccinia Virus Strain Expressing Bacterial Flagellin in Solid Tumor Models"

_viruses, 2023, doi:10.3390/v15040828_

Round 1

Reviewer 1 Report

The paper submitted by Yasmin Shakiba et al describes the construction of recombinant Lister vaccinia viruses deleted for the TK gene, one of which expresses the flagellin of Vibrio vulnificus. The objective is to verify their oncolytic capacities towards 3 types of murine solid tumours (melanoma, colorectal carcinoma, breast cancer) in vitro (B16, CT26 and 4T1 cells) and in vivo after subcutaneous implantation of the cells in immunocompetent mice and treatment by IV. The paper also explores the in vitro viro-induced cytotoxicity in the presence of macrophages, the biodistribution after IV injection of a TK- Fluc + virus to tumour-bearing mice, and the expression of various cytokines in induced tumours or in serum after injection of recombinant viruses. The study is interesting but incomplete and imprecise. Some clarifications are needed and elements need to be corrected.

More specifically:

Materials and methods :

2.2, line 94: indicate the type of 2A peptide used in the constructs

2.2, lines 106-109: specify the number of passages in Rat-2-TK cells in the presence of BrdU, and the number of subcloning by plaque purification cycles. What level of purity was obtained (residual presence of wild-type LIVP virus)? 

2.3: The titration method described in Materials and Methods (quantification in TCID 50 by Reed and Muench method) does not correspond to the one used in the results section (fig 2 and in vivo experiments: titers in PFU). Can you explain this? 

2.9, line 171: specify the dilutions of recombinant viruses used for the test and the corresponding MOI for each of them.

2.10: specify the volumes of virus injected (we only have the titre per ml, so we cannot know the actual quantities injected). Specify the diameters of the tumours at the time of the virus injections.

2.12: only the experiment performed on C57BL/6 mice implanted with B16 cells is shown here, but figure S3 also shows Balb/C mice with 4T1 tumours: complete the materials and methods accordingly.

Results : 

3.1 and figure 1: show also the result of the infection of BHK21 by LIVP-RFP virus; complete the legend (B: ?, C:?, photo on the left, photo in the middle, photo on the right: ?, MOI: ?)

3.2 and figure 2: I do not understand how the viral production is measured (titer in PFU/cell?). This does not correspond to the description in materials and methods. Why is viral production only measured in the supernatant of infected cultures (and not in the cells)? There is only one measurement point (72h): the analysis should be completed by determining the amplification factor (divide the quantity obtained by the initial quantity of the inoculum). As shown in figure 2, CT26 and 4T1 cells are not permissive (to be calculated for B16). It would be interesting to complete the analysis by measuring the genome copy number (qPCR). Furthermore, there is no comparison between cell types (statistical analysis).

3.2 and Figure 3: The level of infection is generally low, especially for LIVP-FlaB-RFP virus. I don't think it can be written that the results show that the recombinant viruses multiply in the tested cancer cells (line 253). Furthermore, it is well stated that there is a significant difference between B16 and CT26 cells infected with LIVP-RFP and LIVP-FlaB-RFP respectively (correct text in lines 254-255).

3.3 and Figure 5: What MOIs were used? Only the results after 48h of co-infection are presented (whereas the 24h and 72h times are also mentioned in the materials and methods): can we have these results? A co-culture control with uninfected cells would also be needed. Correct the legend of figure 5 (replace LIVP-Fluc-RFP by LIVP-RFP).

3.4 and figure 6: Bioluminescence imaging analysis should be complemented by quantification after organ and tumour collection for plaque assay or qPCR at 168h pi

3.5 and Figure 7: There is no evidence that survival is significantly prolonged for treated mice compared to untreated mice except for the B16 treated groups. There is no measurement of the quantity of virus in the tumours at the end of the protocol: this should be corrected (qPCR, or plaque assay).

3.6: the measurement of cytokine quantities seeks to show that flagellin expression would induce a cytokine environment favourable to the recruitment of immune cells in the tumour (macrophages in particular). There is no evidence of this here, and it would have to be verified by a histological study on tumour sections.

small detail: Vaccinia virus is abbreviated as VACV, and the family is called Poxviridae

Reviewer 2 Report

This manuscript constructed recombinant variants encoding bacterial flagellin (subunit B) of Vibrio vulnificus (LIVP-FlaB-RFP), firefly luciferase (LIVP-Fluc-RFP) or red fluorescent protein (LIVP-RFP), and  explored the antitumor efficacy of these variants in syngeneic murine tumor models (B16 melanoma, CT26 colon cancer and 4T1 breast cancer). The results showed that LIVP-FlaB-RFP or LIVP-RFP all mice tumor models exhibited tumor regression with prolonged survival rate in comparison with the control mice. Superior oncolytic activity was observed in the B16 melanoma models treated by LIVP-FlaB-RFP.

However, there are some questions needed to be addressed.

1. It needs to add a title for each figure.

2. In Figure1 B and C, the scale bar is not clear. What is the different between two red fluorescence?  Figure1 D needs to be modified and the two figures should be merged into one.

3. In Figure 2, is the unit of the vertical coordinate Viral production, PFU/cell? If that, the titres of the variants is very low.

4. The capture legend of Figure 6 is not clear. It should describe how to treat the mice including the used virus, mice xenografted with B16.

5. In Figure 7, the results of antitumor effect in mice is not consistent with the cell level in Figure 5D. In Figure 5D, The LIVP-FlaB-RFP showed the obvious killing effect in CT26 and B16 between LIVP-FlaB-RFP and LIVP-RFP even though the authors didn't compare their killing effect. But in Figure 7, the antumor effect in CT26 between LIVP-RFPand LIVP-FlaB-RFP has not significant difference, and the antumor effect in B16 between control and LIVP-RFP has not significant difference. As oncolytic virus, LIVP-REP should show the potent antitumor effect.

6. The authors should discuss the results are not consistent in tumor compared with the serum in Figure 8. 

Reviewer 3 Report

This manuscript describes the insertion of bacterial flagellin into the genome of a TK-deleteted poxvirus strain. Although the approach is promising, it is lacking significant novelty since a very similar publication by Conrad et al has been released in 2015.

In addition, the present manuscript suffers from some essential drawbacks.

1.       The authors should make an effort to convince the reader that their vectors possess superior qualities compared with other very similar ones (Conrad et al., 2015)

2.       The methodology used can be improved or complemented

3.       The in vitro data are not discussed AT ALL in the Discussion section, thus leaving the readers in doubts

4.       The in vivo experimental design is insufficient

-          Histology of the tumor is lacking. In the case of easily accessible tumors like melanoma, this is a major drawback

-          Virus localization in tumors is shown through IVIS; however, virus presence in melanomas should be demonstrated also using histology methods

-          In the Discussion, authors make rather various speculations and describe likely scenarios, but none of them is proven through this paper. For example, they claim that melanoma cell treatment with the flagellin-expressing vector is triggering macrophage activation. This is indeed (not sufficiently) demonstrated in vitro but not in the decisive in vivo experiments

Resubmission may be possible after a significant major revision.

Title

L. 2

Oncolytic efficacy of A recombinant …

Abstract

L. 16

Replace “establishing approach” with e.g. new promising approach, emerging treatment approach/modality

L. 16-17

Oncolytic viruses cause tumor regression through direct cytolysis on the one hand and recruitment and activation of immune cells on the other.

L. 21

Please define IVIS

Introduction

L. 44

…they display AN established high safety profile

L. 46-48

For example, the granulocyte-macrophage colony-stimulating factor (GM-CSF)-expressing recombinant JX-594 is an oncolytic vaccinia virus with increased selectivity for replication in cancer cells

L. 67-68 (and in general throughout the whole manuscript: maybe a native speaker could solve the isuue)

Too many definite articles: Deletion of TK is a common strategy to enhance VV tumor selectivity

Results

L. 248-249

…it has been shown THAT…have HIGHER ability to reproduce in B16…

L. 251

…determined by flow cytometRY…red fluorescenCE or red fluorescent signal

L. 252

Omit ALSO. BHK-21 CELLS were used as a reference

L. 260

Why “interestingly”? Isn’t this selectivity expected for a virus with oncolytic activities?

Figure 2

-          Data on this figure are based on infection at MOI of 1, is this correct? I think you should show also data with MOI 10, for the sake of comparison. Does infection with LIVP-FlaB-RFP (MOI 10) display enough replicative efficiency, in particular in B16 cells?

-          In the Materials and Methods section you state that replication efficiency was determined by the Reed and Muench method. If so, is it correct to use “PFU/cell” (Figure 2 Y axis title)? Why not using a crystal violet stain?

Figure 3

A time point later than 48h makes no sense? Shouldn’t you have at least three time points as long as kinetics is concerned. Moreover, when determining virus cytotoxicity, you do look at 72 hpi. What was the reason of not having this time point by the flow cytometry?

L. 279

Are you sure you really mean “…ENHANCE the direct cytotoxic effect…”? or you rather mean to complement, to synergize with…?

Figure 5

-          Is the vector (presumably produced by the preinfected tumor cells and released into the supernatant) capable of being taken up by the macrophages?

-          Why didn’t you measure some cytokines’ expression in the co-culture supernatant?

Figure 6

If the Fla-expressing viruses are able to infect BHK-21 cells in vitro, isn’t the absence of a luciferase signal in the kidneys strange? How do you interpret this observation

L.315

…it appears THAT

Discussion

L. 351

OncO-selectivity!

L. 354

…virus selectivity for cancer cells

L. 357-358

…the recombinant virus doesn’t infect other permissive cells other than the tumor… Apart from style weakness, this statement does not make sense. If the cells are permissive, why wouldn’t they be infected?

L. 394

… these data may INDICATE…

References

-          Please correct double numbering

-          It would be appropriate if you would cite the article by Conrad et al (J Exp Clin Cancer Res 2015 Feb 19;34(1):19. doi: 10.1186/s13046-015-0131-z.) and devote a short paragraph to it in your Discussion

Round 2

Reviewer 1 Report

Thank you for the answers to the questions and comments and the changes made to the article. Finally, the following points should be modified:

Figure 1 legend, panel B: cut the sentence right after the parenthesis and add a capital letter to Picture

Figure 2: I don't see an asterisk indicating differences between groups, so remove this from the legend

Figure 3: add 72 hours in the legend

lines 403-405: repeat lines 390-392, to be removed

Figure 9: specify the tumor model sampled (B16?) 

Reviewer 2 Report

No more question

Author Response

Thank you.

Reviewer 3 Report

The authors have made a significant effort to improve their article. The present version deserves consideration for publication in Viruses and will certainly represent an interest to the Journal's readership.

Author Response

Thank you.